

# Modeling Biogenic and Anthropogenic Secondary Organic Aerosol in China

Jianlin Hu[1], Peng Wang[2], Qi Ying[1,2], Hongliang Zhang[3], Jianjun Chen[4], Xinlei Ge[1], Xinghua Li[5], Jingkun Jiang[6], Shuxiao Wang[6], Jie Zhang[7,9], Yu Zhao[8,9], Yingyi Zhang[10]

[1] Jiangsu Key Laboratory of Atmospheric Environment Monitoring and Pollution Control, Jiangsu Engineering Technology Research Center of Environmental Cleaning Materials, Jiangsu Collaborative Innovation Center of Atmospheric Environment and Equipment Technology, School of Environmental Science and Engineering, Nanjing University of Information Science & Technology, 219 Ningliu Road, Nanjing 210044, China
[2] Zachry Department of Civil Engineering, Texas A&M University, College Station, TX 77843-3136
[3] Department of Civil and Environmental Engineering, Louisiana State University, Baton Rouge, LA 77803
[4] Air Quality Planning and Science Division, California Air Resources Board, 1001 I Street, Sacramento, CA 95814, USA
[5] School of Space & Environment, Beihang University, Beijing, 100191, China
[6] State Key Joint Laboratory of Environment Simulation and Pollution Control, School of Environment, Tsinghua University, Beijing 100084, China
[7] Jiangsu Provincial Academy of Environmental Science, 176 North Jiangdong Rd., Nanjing, Jiangsu 210036, China
[8] State Key Laboratory of Pollution Control & Resource Reuse and School of the Environment, Nanjing University, 163 Xianlin Ave., Nanjing, Jiangsu 210023, China
[9] Jiangsu Collaborative Innovation Center of Atmospheric Environment and Equipment Technology, Nanjing, Jiangsu 210044, China
[10] School of Environment and Energy, South China University of Technology, Guangzhou, China

Correspondence to: Qi Ying (qying@civil.tamu.edu)

**Abstract.** A revised Community Multiscale Air Quality (CMAQ) model with updated secondary organic aerosol (SOA) yields and more detailed description of SOA formation from isoprene oxidation was applied to study the spatial and temporal distribution of SOA in China in the entire year of 2013. Predicted organic carbon (OC), elemental carbon and volatile organic compounds agreed favorably with observations at several urban areas, although the high OC concentrations in wintertime in Beijing were under-predicted. Predicted summer SOA was generally higher (10-15 µg m$^{-3}$) due to large contributions of isoprene (country average, 61%). Wintertime SOA was slightly lower and was mostly due to emissions of alkane and aromatic compounds (51%). Contributions of monoterpenes SOA were relatively constant (8-10%). Overall, biogenic SOA accounted for approximately 75% of total SOA in summer, 50-60% in autumn and spring, and 24% in winter. Sichuan Basin had the highest predicted SOA concentrations in the country in all seasons, with hourly concentrations up to 50 µg m$^{-3}$. Approximately half of the SOA in all seasons was due to the traditional equilibrium partitioning of semi-volatile components followed by oligomerization, while the remaining SOA was mainly due to reactive surface uptake of isoprene epoxide (5-14%), glyoxal (14-25%) and methylglyoxal (23-28%). Sensitivity analyses showed that formation of SOA from biogenic emissions was significantly enhanced due to anthropogenic emissions. Removing all anthropogenic emissions while keeping the biogenic emissions unchanged led to total SOA concentrations of less than 1 µg m$^{-3}$, which suggests that manmade emissions facilitated



biogenic SOA formation and controlling anthropogenic emissions would result in reduction of both anthropogenic and biogenic SOA.

## 1 Introduction

Fast economic development and rapid industrialization and urbanization in the past several decades not only significantly increase the level of air pollution in China but also lead to higher aerosol loadings in the downwind regions. A significant fraction of the aerosols is made up of organic carbonaceous material (Sun et al., 2014). Globally, organic aerosols (OA) from both biogenic and anthropogenic sources play a significant role in affecting global climate by directly scattering and absorbing solar radiation and indirectly by altering cloud and rain formation processes. It was estimated that China contributed to approximate 16% of global organic aerosol loading in 2000 (Saikawa et al., 2009; Tsigaridis et al., 2005) and its contributions are likely higher nowadays due to increased air pollution levels in recent years. During severe haze events, it was found that secondary organic aerosol (SOA), which formed in the atmosphere from gas-to-particle conversion of primary precursor volatile organic compounds (VOCs), could account for a significant fraction of the total observed organic aerosol (Carlton et al., 2009; Huang et al., 2014a; Tsigaridis and Kanakidou, 2003).

To better quantify the impacts of SOA on human health and climate, and to design more efficient control strategies to improve air quality in China, improved understanding of SOA is urgently needed, including its precursors, formation processes, and physicochemical properties. While a large amount of experimental work, both in the field (Aramandla et al., 2011; Boström et al., 2002) and in the laboratory (Lai et al., 2016), has been carried out in recent years, only a small number of modeling analyses of SOA in China have been reported. In order to determine the contributions of SOA to total organic carbon (OC) burden in China, receptor-oriented analysis with specific SOA tracers (Akagi et al., 2011) or statistical analysis with minimal elemental carbon to OC (EC/OC) ratio (Peng et al., 2013; Wang et al., 2015) have been attempted. Recently, advanced techniques such as the Positive Matrix Factorization analysis of organic aerosol data from Aerosol Mass Spectrometer have also been applied to determine the amount of SOA and its precursors (Huang et al., 2014b; Sun et al., 2014; Wang et al., 2016)

Mechanistic modeling analyses of SOA formation in China have been seldom reported in the literature until recent 5 years or so. The complexity of the models ranges from relatively simple box models (Ciccioli et al., 2014) to full-blown 3D Eulerian models (Hellén et al., 2012; Jiang et al., 2012; Li et al., 2015a; Setyan et al., 2012). Most of the studies focused on the Pearl River Delta (PRD) region in southern China (Ciccioli et al., 2014; Li et al., 2015a; Setyan et al., 2012) and east China (Hellén et al., 2012), and were for short time periods. While computationally intensive, a full-year simulation for the entire China region with a fully coupled meteorology-chemistry model was also attempted (Jiang et al., 2012).





The SOA modules used in these studies applied the relatively simple traditional two-product approach (Ciccioli et al., 2014) or the more complex volatility basis set (VBS) approach to better represent multi-generation oxidation of SOA precursors in the gas phase and its aging processes in the aerosol phase (Hellén et al., 2012). The two-product approach has been shown to under-predict SOA in many previous studies. While the VBS approach and many of its variants have been show to predict

higher SOA concentrations, a recent study showed that the VBS parameters adjusted to fit individual SOA precursor well in photochemical chamber experiments cannot produce satisfying results to simulate chamber experiments with mixed precursors (US EPA, 2004). In addition, both two-product and VBS approach assume equilibrium partitioning of semi-volatile products, however, recent studies have revealed that reactions on particle surfaces or with particle water under acidic conditions of certain compounds such as dicarbonyls (e.g. glyoxal) and epoxydiols from isoprene lead to rapid formation of low volatile

oligomers, organo-sulfate and organo-nitrates (George et al., 2015). The heterogeneous SOA formation pathway from aqueous uptake of glyoxal (GLY) has been studied for the PRD region, and it was found that including glyoxal surface uptake leads to higher isoprene SOA (iSOA) fraction (Li et al., 2015a).

Isoprene epoxydiols (IEPOX) and methacrylic acid epoxide (MAE) formed from isoprene oxidation under low and high NOx

conditions (Lin et al., 2013; Surratt et al., 2010), respectively, have been reported to form low-volatile organic products in aerosol water under strong acidic conditions (Karambelas et al., 2014; Pye et al., 2013). Some of the recent SOA modeling studies included more detailed treatment of isoprene oxidation in the gas phase and updated SOA modules by using either a surface-controlled uptake approach (Li et al., 2015b; Ying et al., 2015) or a more mechanistic approach for the aqueous reactions (Pye et al., 2013). It was found that IEPOX accounted for as much as 34% of the total iSOA in summer (Ying et al.,

2015). Most of these modeling studies were for North America, and whether these processes are important to SOA formation in China remains unclear.

In this study, we simulated ground level SOA concentrations in China during the entire year of 2013 using a regional 3D air quality model. The model includes treatment of the isoprene gas phase chemistry that leads to the production of IEPOX and

MAE, and updated SOA mechanism with updated two-product yields and SOA formation from reactive surface uptake of dicarbonyls and IEPOX and MAE. This is the first time a regional SOA model with recent advances in iSOA treatment has been applied in China. This study provides new insights into the importance of biogenic and anthropogenic emissions on SOA formation in China under different meteorological and emission conditions, a better evaluation of the current levels of SOA concentrations and the contributions of anthropogenic and biogenic sources to SOA loading throughout the country.



## 2 Method

### 2.1 Model description

The Community Multiscale Air Quality (CMAQ) model developed by the United States Environmental Protection Agency (US EPA) with an updated gas phase photochemical mechanism and a revised SOA module was applied in this study. A

complete description of the SOA module has been described in greater details in Ying et al. (2015), and a short summary is provided below. The CMAQ model was also updated to include heterogeneous reactions of $NO_2$ and $SO_2$ on particle surface to form secondary nitrate and sulfate (Ying et al., 2014).

The gas phase photochemical mechanism applied in this study is based on the SAPRC11 (S11) photochemical mechanism

(Carter and Heo, 2012), an update of the widely used SAPRC07 (S07) mechanism. The S11 mechanism was further modified to include a more detailed isoprene oxidation scheme to predict the formation of IEPOX and MAE, as originally described by Xie et al. (2013) and Lin et al. (2013). In addition, this modified S11 mechanism was designed to track GLY and methylglyoxal (MGLY) from six major groups of precursors using a tagged-species approach. The six groups of precursors are: (1) aromatics with OH reaction rates less than $2 \times 10^4$ ppm$^{-1}$ min$^{-1}$ (ARO1), (2) aromatics with OH reaction rates greater than $2 \times 10^4$ ppm$^{-1}$

min$^{-1}$ (ARO2), (3) isoprene (ISOP), (4) monoterpenes (TERP), (5) sesquiterpenes (SESQ), and (6) primary emissions of GLY and MGLY.

Two major SOA formation pathways were included in the SOA module. The first is the equilibrium partitioning of semi-volatile products from oxidation of long-chain alkanes (ALK5), ARO1, ARO2, ISOP, TERP and SESQ, and subsequent

formation of non-volatile oligomers based on the traditional two-product approach as implemented in the version 6 of the aerosol module in CMAQ (AERO6). Oligomers from different precursors were also treated using the precursor-tagging approach so that a complete analysis of precursor contributions to SOA can be conducted. In this study, the SOA mass yields were updated from the ones used in Ying et al. (2014) and Ying et al. (2015) to account for vapor wall loss during chamber experiments based on data provided by Zhang et al. (2014). The original and updated mass yields are listed in Table 1. The

second SOA formation pathway in the model is the surface controlled irreversible reactive uptake of GLY, MGLY, IEPOX and MAE. Uptake coefficients (γ) for GLY and MGLY were fixed at $2.9 \times 10^{-3}$ taken from Fu et al. (2008) and an acidity dependent uptake coefficient for IEPOX and MAE was based on the parameterization described by Li et al. (2015b). The resulted $\gamma_{isoepox}$ in the summertime (June, July and August) in this study are approximately $4-10 \times 10^{-4}$ in most areas (see Fig. S1). These $\gamma_{isoepox}$ values are in the range mentioned by Pye et al. (2013), which were based on more mechanistic calculations.

The isoprene SOA yields in the CMAQ model were taken from Kroll et al. (2006), which was conducted under dry conditions (RH<10%). Thus the yields derived from the experiments do not include contributions from reactive surface uptake, which only occurs on wet particles.



## 2.2 Model Application

The modified CMAQ model was applied to simulate surface concentrations and determine precursor contributions to SOA in China for the entire year of 2013. The model domain covers China and surrounding areas in East and Southeast Asia, with a horizontal resolution of 36 km (see Fig. S2). The meteorology inputs needed to drive the air quality simulations were generated by the Weather Research and Forecasting (WRF) model v3.6.1 using the NCEP FNL Operational Model Global Tropospheric Analyses dataset for boundary and initial conditions. The anthropogenic emissions were based on the Multi-resolution Emission Inventory for China (MEIC) (http://www.meicmodel.org) with a resolution of 0.25º × 0.25º resolution and a matching speciation of VOCs for the SAPRC mechanism. Biogenic emissions were generated using the Model for Emissions of Gases and Aerosols from Nature (MEGAN) v2.1 with year-specific leaf area index (LAI) data from the 8-day Moderate Resolution Imaging Spectroradiometer (MODIS) LAI product (MOD15A2). The Fire INventory from NCAR (FINN) was used for open biomass burning emissions (Wiedinmyer et al., 2011). Seasonal average emissions of SOA precursors (ALK5, ARO1, ARO2, ISOP, TERP and SESQ) are shown in Fig. S3. Inline processing was used to generate dust and sea salt emissions during CMAQ simulations. More details of the emission processing and meteorology model result evaluation can be found in Hu et al. (2016).

## 3 Results

### 3.1 Model evaluation

The capability of the model in predicting ozone and $PM_{2.5}$ mass concentrations has already been evaluated by comparing with observations at 60 major urban areas and described in much greater details in Hu et al. (2016). In summary, model performance of ozone (1-hour peak and 8-hour average) and $PM_{2.5}$ mass concentrations (daily average) was considered generally within the model performance criteria recommended by US EPA. Model performance is generally better in the more economically developed regions such as YRD and North China Plain (NCP), because of more accurate emission inventories than the less developed regions such as the northwest. Nonetheless, as ozone and a significant portion of the $PM_{2.5}$ mass are formed from photochemical reactions of the precursors, the general capability of the model in reproducing these species suggests that the oxidation capability of the atmosphere and precursor concentrations essential for SOA predictions were reasonably predicted – even though the SOA formation processes are much complex and less understood.

In the following, predicted concentrations of several VOCs, as well as OC and EC are compared with available observations during this period in order to provide more supporting evidence for the SOA estimations.



### 3.1.1 Volatile organic compounds

Predicted hourly concentrations of lumped primary VOCs (ARO1, ARO2 and ALK5) and ISOP which act as SOA precursors in the S11 mechanism are compared with observations made at Nanjing University of Information Science and Technology (NUIST) during August 2013 and shown in Figure 1. In addition to these SOA precursors, predicted ethene (ETHE), lumped

primary olefins (OLE1 and OLE2), acetaldehyde (CCHO) and lumped higher aldehydes (RCHO), which have significant contributions to the oxidation capacity of the atmosphere are also compared with observations. While both CCHO and RCHO can be directly emitted, in the polluted urban areas most of them are formed from oxidation of other VOCs. Also included in the comparison were MACR and MVK, oxidation products from isoprene. Methyl-ethyl-ketone (MEK), which can be both emitted as primary species as well as formed secondarily from oxidation of a number of VOCs and their oxidation products

(such as MACR and MVK) are also included in the evaluation. The observed concentrations of 54 detailed VOC species are used to calculate the concentrations of the lump S11 species. The VOC monitoring site is located on the rooftop of the Science Building on campus of NUIST. Measurement of these VOC species followed the same protocol as those used in the Photochemical Air Monitoring Station (PAMS) of US EPA. The predicted concentrations at the grid cell where the monitor is located are used to compare with observations. To illustrate the spatial variability of the VOC species, the ranges of the VOC

concentrations in the nine grid cells with the monitor site at the center are also shown in the comparison.

As shown in Figure 1, predicted anthropogenic primary emitted species (ETHE, OLE1, OLE2, ARO1, ARO2 and ALK5) have large spatial variations. This indicates that their emissions are likely much higher in urban areas but significantly lower in rural areas. Day-to-day variations of these species are generally well captured. For example, the decrease of ETHE as well as other

primary species during August 20-24 and the subsequent rapid increase on August 25 are correctly predicted. This day-to-day variation is due to changes in the meteorological conditions. Emissions remain constant in the same month except for the weekday/weekend variations, and other factors that cause day-to-day variations of emissions (such as plume rise) within the same month were not considered. Observed OLE2 concentrations show erratically high concentrations during August 20-22, which are likely due to impact of the local emission near the sampling site or measurement errors. For ARO1 and ARO2, the

two major SOA precursor, predicted concentrations at the sampling site are slightly lower than observations (mean fractional bias, MFB=-0.63 and -0.77, respectively). ALK5 concentrations are lower than aromatic compounds, and show both over- and under-predictions, with an overall MFB of -0.05 and mean fractional error (MFE) of 0.69. As the olefin species are important in affecting the oxidation capacity of the atmosphere, the good agreement between model predictions and observations suggests that the initial oxidation rate of the SOA precursors can be reasonably predicted at this location.

Predicted isoprene concentrations show distinct day-to-day variations, with higher concentrations (up to approximately 8 ppb at peak hours) in the first half of August and lower in the second half (up to 1-2 ppb in the afternoon). The predicted high concentration peaks are sharp, usually last for 1-2 hours only. Observed isoprene concentrations do not show significant day-



to-day variations, with peak-hour concentration of approximately 2 ppb throughout the month. As isoprene is a short-lived species under typical day-time urban atmospheric conditions, the predicted high concentrations could not be due to regional transport for the large grid cell size used in this study. Variations in the predicted day-to-day concentrations must be due to isoprene emissions estimated by the MEGAN model. A previous study indicated that the MEGAN model might have

overestimated the amount of isoprene emission vegetation in the urban areas due to incorrect vegetation types and fractions and leaf area index (Kota et al., 2015). However, a few other studies suggested that a significant portion of the observed isoprene was not from live vegetation but from other sources such as vehicles in urban areas (Borbon et al., 2001; Hellén et al., 2012; McLaren et al., 1996). More detailed analyses are needed to determine what triggered the rapid increase in the predicted emissions of isoprene in the afternoon.

Oxidation products of isoprene, MACR and MVK, show better agreement between predictions and observations (MFB=-0.14 and -0.35, and MFE=0.68 and 0.74, respectively). Observed concentrations of these species are actually higher on the first half of the month, which can cast some doubt on the isoprene measurements. However, these species are much longer-lived than isoprene, therefore transport from high isoprene emissions regions can contribute to the higher observed concentrations. In the

second half of the month, regional emissions of isoprene decrease due to changed weather conditions, and thus reduce the amount of MACR and MVK transported to the urban grid cell where the monitor is located. In summary, good agreement of these two species with observations provides a good starting point for iSOA predictions.

For the other oxidation products (CCHO, RCHO and MEK), RCHO and MEK show excellent agreement between observed

and predicted concentrations (MFE=-0.04 and 0.16, and MFE=0.68 and 0.53, respectively). CCHO observations are less complete during the month and show much higher concentrations than predictions from August 14-21. For all three species, a large peak in the predictions on August 11 is observed. However, VOC observations were not available because of an equipment problem on August 10 and 11, making it impossible to evaluate if these predicted peaks actually occurred or not. Although this evaluation of VOC species is only for a single location in a single month due to limited observation data available

during the studying period, it provides support that the VOC emissions and chemistry, at least in this region, are reasonably represented by the modeling system.

### 3.1.2 Elemental and organic carbon

Predicted EC and OC concentrations are also compared with observations, as shown in Figure 2. EC and OC measurements

are available at serval locations: two locations in Beijing (on the campuses of Tsinghua University (Cao et al., 2014) and Beihang University (Wang et al., 2015), January and March 2013), one in Nanjing (Jiangsu Provincial Academy of Environmental Science (Li et al., 2015a), December 2013) and one in Guangzhou (Tianhu, January and February 2013 (Lai et al., 2016)). As shown in Figure S2, Nanjing is located in eastern China near the YRD, approximately 1000 km to the south of





Beijing. Guangzhou is a mega city located in southeast China near Hong Kong. These three cities represent very different climate and emission conditions, and likely have different SOA formation pathways. Daily average concentrations were measured with different sampling intervals.

Figure 2(a) and Figure 2(b) show predicted and observed OC concentrations in Beijing in January 2013, the month characterized by several high PM pollution episodes that covered large parts of the country. The Tsinghua and Beihang campuses are 3 km apart and are located in the same model grid cell. Observation data are available at the Tsinghua site only during January 8-14, and are very similar to the observed concentrations at the Beihang site. The close agreement of the two sets of measurements during high OC days suggests that the high concentrations (as high as 60-80 μg m$^{-3}$) are not due to local
sources very close to monitors but likely a regional phenomenon. EC predictions agree well with observations (Figure 2b; MFB=0.23 and MFE=0.48). While the model predictions of OC generally agree with the observations for day-to-day variations (MFB=-0.36, MFE=0.53), they are significantly lower than observations on high concentration days, except the last pollution episode in the last week of January. The good agreement of EC even during high pollution days suggests that primary emissions and meteorological fields are accurate enough and are not the major cause of OC under-prediction. Therefore, the OC under-
prediction is more likely due to under-prediction of SOA from anthropogenic sources during high pollution days.

Figure 2(c) and Figure 2(d) show that day-to-day variations in the OC and EC concentrations in Beijing during March 2013 are also well captured (MFB=-0.59 and -0.17, and MFE=0.61 and 0.42). Spatial variations of the OC concentrations are much smaller than those in January, as primary emissions of OC from the urban areas are much reduced. OC is still more under-
predicted than EC, except for the first week of March when both OC and EC concentrations are under-predicted greatly, which suggests missing primary emissions such as open burning. Figures 2(e) and 2(f) show that OC and EC concentrations in Nanjing during December 2013 are predicted well with slight under-prediction (MFB=-0.24 and -0.24, MFE=0.39 and 0.31, respectively). The day-to-day variations are also well captured with the predicted OC and EC concentrations reaching 30 and 10 μg m$^{-3}$ with small spatial variations. The under-prediction of OC is, at least partially, due to under-prediction in primary
emissions, as both EC and OC concentrations are under-predicted. Figures 2(g-j) show the comparison of EC and OC in Guangzhou during January and February, 2013. Although both OC and EC were in reasonable agreement with the observations on the days when the observations are available (MFB=-0.05 and 0.48, and MFE=0.33 and 0.48, respectively), it is hard to conclude whether the day-to-day variations are correctly captured since observations were made every six days.

In summary, available OC and EC observations in winter and spring in three major cities suggest that emissions of EC, and thus possibly emissions of primary OC (POC), are well estimated. POC provides the medium for the partitioning of SVOCs formed from oxidation of the precursors, and correctly predicting POC is necessary for correctly predicting SOA concentrations. There are also indications that SOA are likely under-predicted, especially during the high pollution days in winter in Beijing.





## 3.2 Regional distribution of SOA

### 3.2.1 Mass concentrations of SOA

Figure 3 shows predicted seasonal average SOA concentrations in (a) spring (March, April and May), (b) summer (June, July and August), (c) autumn (September, October and November), and (d) winter (December, January and February) of 2013. In spring, predicted SOA concentrations peak in Southeast Asia, which is associated with a larger amount of open burning emissions, as well as higher temperature and stronger solar radation that enhance the photochemical production of SOA. Some SOA is transported to southwest China by the southeast monsoon, which contributes to high SOA concentrations (~20 μg m$^{-3}$) in Yunnan and part of Guangxi provinces. These high SOA regions are ideal for further studies to better understand the impact of large scale open burning on SOA and air quality. Relatively high SOA concentrations are also predicted in other southern provinces and in the Sichuan Basin, with seasonal average concentrations reaching ~10 μg m$^{-3}$. Predicted SOA concentrations are low in the north and northeast provinces. As indicated in Figure S4(a), SOA accounts for approximately 20-50% of OA in majority of the areas where SOA concentrations are high.

In summer, SOA concentrations are generally more than 5 μg m$^{-3}$ in a large portion of the populated areas in central and eastern China. Central provinces, such as Jiangsu, Henan, Shandong and the southern part of the Hebei provinces in the NCP have concentrations up to 10-15 μg m$^{-3}$. This shift of high SOA regions from southeast to central China is accompanied by the change in the average surface wind direction from erratic wind in spring to generally southly wind in summer. Figure S4(b) shows that SOA is the major form of OA in summer, accounting for approximately 50-70% of the OA in the densely populated areas and more than 80% in most other areas.

Figure 3(c) shows that as the wind direction changes to mostly northly again in autumn, higher SOA concentration regions move south. The highest concentration in autumn is approximately 8 μg m$^{-3}$. The spatial distribution of SOA in autumn is similar to that in spring, although lacking of precursor emissions and transport of SOA from southeast Asia leads to lower SOA concentrations in the south border provinces. Relative contributions of SOA to total OA in autumn is approxiamtely 30-50%, which is also similar to that in spring, as shown in Figure S4(c). Interestingly, model predictions show that SOA concentrations in winter can still be high, with highest seasonal average concentrations reaching approximately 20 μg m$^{-3}$ in the Sichuan Basin and 12-14 μg m$^{-3}$ in the Henan province in central China. The rest of the populated areas in central and eastern China have concentrations in the range of 5-6 μg m$^{-3}$. While the SOA still accounts for a noticable fraction of OA in east and southeast China, its contributions to total OA is only approximately 10% in north and northeast China (Figure S4(d)) due to large emissions of POA from residential sources (Hu et al., 2015). Predicted SOA concentrations in Xinjiang, Tibet and some other west and northwest provinces are always low due to low emissions of precursors (Hu et al., 2016). Averaged





over the entire year of 2013, as shown in Figure S5, it is predicted that average SOA contribution to total OA is approximately 30%, and the highest SOA concentration in China is in the Sichuan Basin with an annual average concenteration of 10-12 μg m$^{-3}$.

**3.2.2 SOA from traditional and reactive surface uptake pathways**

Figure 4 shows the predicted seasonal average concentrations of SOA components, which represent SOA formation from different pathways: equilibrium partitioning of semi-volatile SOA (SEMI), oligomerization of the condensed semi-volaitle SOA (OLGM), and surface irreversible uptake of IEPOX, MAE, GLY and MGLY. The SEMI and OLGM pathways are considered as traditional because they are included in the default CMAQ aerosol module AERO6. The SEMI contributions to

SOA are on the order of 2 μg m$^{-3}$, and did not vary significantly in different seasons. The sptial distribution of the SEMI components generally follows the same distribution as the total SOA but shows more spatial heterogenity, with higher concentrations near the source regions. This is because the primary precursors generally become oxidized relatively quickly after they are emitted. In the current SOA module, oxidation of primary precursors leads to immediate formation of two semi-volatile products, which then partition to the particle phase assuming instant equilibrium. In contrast, the OLGM components

have wider and more smooth spatial disitribution, as they are formed slowly from the SEMI components. In the current parameterization, the SEMI components turn into OLGM following a first order reaction with a half-life of 20 hours. The contributions of OLGM to total SOA are higher in spring and winter (~27%, see Table 2) and lower in summer and autumn (21-23%), even though the relative contributions of the SEMI components to total SOA remain relatively constant (~20%) in all seasons. This is likely due to more efficient removal such as wet deposition in the warmer months. Overall the traditional

pathways account for 43-47% of the total SOA. This is higher than summertime SOA simulations for the eastern US, where SOA from the traditional pathways is approximately 16% because the contributions of isoprene to total SOA is much higher (Ying et al., 2015).

Contributions of MAE to total SOA is neglibible (<1%). It is known that MAE is typically formed under high NOx conditions,

but isoprene emission is generally high in rural and remote areas where NOx emissions are low. Even though occationally transport might mix urban plumes with sufficient isoprene, creating a high NOx environment locally, this did not occur frequently enough to make significant contributions. SOA from IEPOX shows clear spatial and temporal distributions. IEPOX is predicted to be a major contributor to iSOA, with contributions of ~13-14% in spring while lower in autumn (~9%) and winter (~5%), averaged over the entire country. In some downwind regions, the contributions of IEPOX are much higher. GLY

and MGLY have contributions from both primary emissions and secondary formation. Spatial distributions of MGLY are wider than those of GLY, suggesting that more MGLY is due to secondary formation. Seasonal variations of GLY are also diffierent from those of MGLY. While contributions of GLY to SOA are highest in winter (~25%) and much lower in summer (~14%), contributions of MGLY are relatively constant, with slightly higher contributions in summer (~28%) and lower in



winter (~23%). Emissions in winter increase significantly in China, mostly due to residential heating, suggesting that the importance of anthropogenic GLY emissions in China.

### 3.2.3 SOA from different precursors

Figure 5 shows the predicted contributions to seasonal average SOA concentrations due to different types of precursor species: aromatic compounds (ARO=ARO1+ARO2), ALK5, isoprene, TERP, SESQ and primary glyoxal and methylglyoxal ([M]GLY=GLY+MGLY). In spring, isoprene is the largest contributor to total SOA, and explains most of the spatial distribution of SOA shown in Figure 3(a). The peak contributions of 4-5 $\mu$g m$^{-3}$ occur in Sichuan Basin and south China. Figure 3(a) indicates some border provinces experienced cross-border transport of SOA from neighboring countries. Isoprene

SOA in these near-border regions are as high as 10 $\mu$g m$^{-3}$. While it has been reported that large amounts of isoprene and other biogenic VOCs can be emitted from biomass burning (Ciccioli et al., 2014), the emission factors of isoprene vary significantly for different burning activities and fuel types (Akagi et al., 2011). Thus, the large amount of isoprene SOA predicted from open burning in this region needs to be confirmed by additional experimental studies to quantify the sources, by detailed analysis of SOA tracer compounds, for example. Averaged over the entire region in China, isoprene contributes to

approximately 48% of the total SOA in spring (see Table 3), and TERP contributes to 10% (~2 $\mu$g m$^{-3}$). TERP contributions are also confined in the forested regions in south China. ARO accounts for 21% of the total SOA in spring, and its spatial distribution is much wider, contributing a relatively uniformed concentration of approximately 2 $\mu$g m$^{-3}$ throughout the central, east, south and southeast China.

Isoprene SOA also dominates the total SOA predicted in summer, with the highest concentrations in the Sichuan Basin. High iSOA regions cover the NCP, the three provinces in the northeast and some provinces in east and central China. County-average contributions of iSOA account for 61% of total SOA. Relative contributions of isoprene to total SOA is shown in Figure S6(b). Most of the areas in central China and NCP do not have significant emissions of isoprene. With summer southerly wind, significant regional transport of iSOA from south China contributes to the PM2.5 mass loading in the downwind central

China and NCP areas. In contrast, SOA from TERP and SESQ does not show as much regional transport, and is generally located in southeast China where their emissions are highest (see Figure S6(c) and S6(d), and Figure S3). This is because GLY and MGLY from TERP and SESQ are much lower than from isoprene. Detailed analysis of different formation pathways to iSOA is discussed in Section 4.2. Together with contributions from TERP (10%) and SESQ (4%), the estimated contribution of biogenic SOA to total SOA in summer is approximately 75%. Increases in the solar radiation and temperature in the summer

facilitates the biogenic emissions. Absolute contribution of ARO to SOA in summer is approximately 2 $\mu$g m$^{-3}$ in NCP, similar to that in spring. The high fractional contribution of SOA from ARO (~25%) is in the coastal regions in east China where the ARO emissions are high. However, the country-average relative contribution of ARO to total SOA decreases from 21% in spring to 14%, due to significant increase in biogenic SOA,



In autumn, contributions of isoprene and monoterpene SOA decrease rapidly. The spatial disitributions are similar to those in spring, but without high SOA in the southern border provinces. Country-average biogenic contributions to SOA decrease to approximately 36%. In winter, contribuiton of ARO increase in Sichuan Basin and central China, reaching peak concentrations of 4-5 µg m$^{-3}$. This leads to a significant increase of its country-average contributions from 14% in summer to 31% in winter. The contributions of biogenic emissions to SOA is small in most areas with contributions less than 1 µg m$^{-3}$. Contributions of primary GLY and MGLY increase signficantly, up to 6 µg m$^{-3}$ in Sichuan Basin and in central China. The country-average contribution increases from 7% in summer, 11-15% in spring/autumn to 25% in winter. The spatial distribution patterns of primary GLY and MGLY SOA do not vary much among seasons. As GLY and MGLY form SOA directly through irreversible surface uptake reactions, and there are always sufficient particles in the polluted atmosphere, this suggests that once the SOA are formed on particles, they are not transported over long distances as effectively as gas precursors and their intermediate oxidation products. The timescale of the gas phase multistep oxidation reactions can greatly affect the long range transport and thus spatial distribuiton of SOA. Residential sources are the sole important contributor to total primary GLY and MGLY emissions in China (see Figure S7). The large contributions of primary GLY and MGLY to SOA formation were not found in the previous study in eastern US as residential emissions were not significant (Ying et al., 2015).

### 3.3 Time series at different cities

Figure 6 illustrates the time series of SOA in January and August of 2013 at four cities, i.e., Beijing in NCP, Shanghai in YRD, Guangzhou in PRD and Chengdu in Sichuan Basin. The four cities are located in the four most economically developed regions and have quite different emissions and meteorological conditions. In all cities, summertime (August) SOA concentrations are higher than wintertime (January). Chengdu has experienced the highest SOA concentrations among the four cities both in winter and summer, with the highest hourly concentrations of 35 µg m$^{-3}$ and 50 µg m$^{-3}$ in January and August, respectively. Peak hourly SOA concentrations in other cities in January are approximately 20 µg m$^{-3}$. In August, SOA concentrations in Shanghai and Beijing could reach approximately 40 µg m$^{-3}$. Aromatic compounds are important in winter, with contributions as much as 10 µg m$^{-3}$ on high SOA days in all cities. Primary GLY and MGLY are also important in winter, especially in Chengdu. In summer, isoprene is the most important precursor of SOA in all cities. However, the contributions of ALK, ARO and [M]GLY are not negligible.





## 4 Discussion

### 4.1 SOA from primary and secondary glyoxal and methylglyoxal

As the reversible uptake of GLY and MGLY plays an important role in SOA formation, the precursors that produce these two species are investigated using GLY and MGLY results from the precursor-tracking model. As disucssed in section 3.2.2, there are significant seasonal variations in the precursor contributions. Tables S1 and S2 show that isoprene is the most important secondary source of GLY and MGLY SOA in summer, on average accounting for 53% and 85% of total GLY and MGLY SOA, respectively. This agrees well with a previous study indicating that 47% and 79% of global gas phase glyxoal and methylglyoxal were from isoprene oxidation (Fu et al., 2008). ARO accounts for 12% of GLY and 11% of MGLY SOA in summer, and its contribution increases to 31% of GLY and 52% of MGLY SOA in winter. Contribution of primary GLY accounts for as much as 61% of the total GLY SOA in winter, while the contribution of secondary GLY is 65% in summer. As described in Section 3.2.1 and 3.2.2, most of the MGLY is secondary, contributing from 96% of the total MGLY SOA in summer to 74% in winter. Liu et al. (2016) compared gas phase GLY predictions with satellite observations and found that GLY was underestimated and an increase of aromatic emissions by 4-10 times was needed to bring the predicted GLY in line with the observations. Such a large under-estimation of aromatic compounds is not found in this study based on the observation data in Nanjing, although primary GLY is possibly underestimated. As the primary GLY emissions from residential sources have large seasonal variations, more detailed analyses of the predicted glyxoal with observations both at the surface and in space, at different seasons, are needed to futher constrain the emissions of GLY and its precursors.

### 4.2 Isoprene SOA formation pathways

Analyses in Section 3 show that contribution of isoprene to total SOA is high in spring and summer. A more detailed analysis of the iSOA was conducted to reveal the contributions of components from different pathways to the predicted summertime iSOA concentrations, as shown in Figure 7. Averaged over the entire country, SEMI and OLGM component account for only 27% of the predicted iSOA. The rest of the iSOA is due to GLY (10%), MGLY (35%), IEPOX (27%) and MAE (1%). Spatial distributions of these components are shown in Figures 7(b)-7(e). The SEMI component is high in the Sichuan Basin, central and east China, close to the forest regions of south China with high isoprene emissions. The other components show a generally similar spatial pattern with high concentrations in the Sichuan Basin, NCP, and YRD regions.

### 4.3 Sensitivity of predicted SOA to isoprene/NOx/VOCs emission changes

The formation of SOA is affected by both biogenic and anthropogenic emissions. To evaluate the changes in SOA concentrations with respect to changes in anthropogenic and biogenic emissions, multiple sensitivity simulations for SOA in January and August 2013 were conducted, as summarized in Table 4. The 50% reduction in NOx emissions and 25% reduction





in VOC emissions used in simulations 2-4 represent a rough estimation of the emission reduction target for the year 2020 based on data from Chatani et al. (2014) for a stringent emission control scenario. In simulation 5, biogenic emission was reduced by 50% while anthropogenic emissions remained unchanged, as some studies showed that the MEGAN model might overestimated biogenic emissions (Carlton and Baker, 2011). This was used to evaluate the sensitivity of SOA predictions on

biogenic emissions. Recently, studies of biogenic SOA in southeast United States suggest that SOA from isoprene could be enhanced by anthropogenic emissions thus reducing anthropogenic emissions might have extra benefit in reducing both biogenic and anthropogenic SOA (Carlton et al., 2010). Simulation 6 was designed to test the magnitude of this interaction in China by removing all anthropogenic emissions.

Figure 8(b) and 9(b) show that reducing NOx emissions alone by 50% would result in both increase and decrease of predicted SOA. In both seasons, decrease of SOA by approximately 1 $\mu$g m$^{-3}$ occurs in the south and in the Sichuan Basin while increase of SOA of 1-2 $\mu$g m$^{-3}$ occurs in the Beijing-Tianjin-Hebei (BTH) and YRD regions. However, the causes of the changes are different. Analysis of the SOA components in August showed that the increase of SOA in the domain is mostly due to increase in the SOA from the IEPOX pathway (Figure S8). The decrease of SOA is mostly due to reduction in the SOA from GLY and

MGLY, which are mainly because of reduction of GLY and MGLY from isoprene (compare Figure S8 and Figure S9). In January, the increase and decrease in the SOA are due to changes in the SOA from ARO and ALK5 (Figure S10(d,e)). SOA from primary GLY and MGLY (Figure S10(f)) also showed similar spatial patterns. In the current mechanism, ARO, ALK5 and GLY/MGLY only react with OH to form semi-volatile SOA and oligomers. The result suggests that the spatial pattern of the SOA changes in winter due to NOx is likely due to increase of the OH concentrations associated with NOx reduction in

the high NOx emission regions. Increase of SOA also occurs in regions downwind of the mega cities due to regional transport, even though these downwind regions may not be NOx-saturated based on local emissions.

As shown in Figure 8(c), reducing anthropogenic VOC emissions by 25% in August results in a decrease of SOA throughout the domain by 1-1.4 $\mu$g m$^{-3}$, or a relative change of approximately 10%. In January, 25% reduction of VOCs leads to

approximately 20% reduction of the predicted SOA concentrations, or up to 3 $\mu$g m$^{-3}$ (Figure 9c). This higher amount of reduction is expected because most of the SOA in winter is due to anthropogenic emissions. Figure 8(d) shows that simultaneous reduction of NOx and VOC in August is more effective in reducing SOA concentrations than NOx or VOC emission reduction alone. However, Figure 9(d) shows that in January reducing both led to slightly less SOA reduction than reducing VOC alone. As shown in Figure 9(e), decrease of isoprene by 50% results in decrease of total SOA by as much as 4

$\mu$g m$^{-3}$ (or 35%) in August. Analysis of the SOA components shows that both iSOA and SOA from other precursors are decreased. This result highlights that the accuracy of the estimation of the biogenic isoprene and other biogenic VOCs can greatly affect the model predictions of iSOA. Given the high sensitivity of predicted SOA and iSOA to isoprene emissions, more studies should be carried out to evaluate, constrain and refine biogenic emission estimations in China.





Removing all anthropogenic emissions while keeping the biogenic emissions unchanged (including soil NOx) leads to very low SOA concentrations, even in the summer. Most areas are predicted to have SOA concentrations less than 2 µg m$^{-3}$ in summer and less than 1 µg m$^{-3}$ in winter. Since the reported dependence of iSOA formation due to particulate nitrate and

sulfate (Xu et al., 2014) was not modeled in the current study, the significant reduction of iSOA in the current study is due to reduced oxidation capacity of the atmosphere (i.e. lower OH during the day and NO$_3$ radical at night) that leads to slower formation of semi-volatile and oligomers as well as lower acidity of the aerosols that reduces the uptake coefficient of IEPOX thus less SOA. None the less, the results from sensitivity simulation 6 clearly demonstrates that human related emissions are actually responsible for almost all SOA. Dividing the SOA concentrations into biogenic and anthropogenic portions by the

carbon source alone will erroneously under-estimate the impact of manmade emissions on aerosol loadings (Setyan et al., 2014; Setyan et al., 2012).

## 5 Conclusions

In China, predicted SOA concentrations are generally higher in summer (10-15 µg m$^{-3}$) due to large contributions of isoprene

(country average, 61%) and lower in wintertime due to emissions of alkane and aromatic compounds (51%). Overall, 75% of total SOA in summer, 50-60% in autumn and spring, and approximately 24% in winter are due to biogenic SOA. The highest SOA in mainland China occurs in the Sichuan Basin in all seasons, with hourly concentrations as high as 50 µg m$^{-3}$. Regarding the SOA formation pathways, approximately half of the SOA in all seasons is due to the traditional equilibrium partitioning of semi-volatile components followed by oligomerization, while the remaining is mainly due to reactive surface uptake of

isoprene epoxide (IEPOX, 5-14%), glyoxal (GLY, 14-25%) and methylglyoxal (MGLY, 23-28%). Manmade emissions facilitate biogenic SOA formation and controlling anthropogenic emissions would result in reduction of both anthropogenic and biogenic SOA.

**Acknowledgement**

The authors J.H., X.G. and X.L. would like to thank the support from the National Natural Science Foundation of China (91544220 and 41275121), Natural Science Foundation of Jiangsu Province (2151071501901), Jiangsu Specially-Appointed Professor Project (2191071503201), and the Priority Academic Program Development of Jiangsu Higher Education Institutions (PAPD), and Jiangsu Province Innovation Platform for Superiority Subject of Environmental Science and Engineering (No. KHK1201). The authors also want to acknowledge the Texas A&M Supercomputing Facility

(http://sc.tamu.edu/) for providing computing resources useful in conducting the research reported in this paper.





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



**Table 1 Updated aerosol mass yields used in this study that account for the wall-loss using scaling factors based on Zhang et al. (2014)**

|      | Updated |        | Original* |        |
|------|---------|--------|-----------|--------|
|      | $\alpha_1$ | $\alpha_2$ | $\alpha_1$ | $\alpha_2$ |
| ALK  | 0.0865  | -      | 0.0865    | -      |
| ARO1 | 0.2545  | 0.7623 | 0.2253    | 0.6764 |
| ARO2 | 0.2545  | 0.7653 | 0.2253    | 0.6764 |
| TERP | 0.1811  | 0.5905 | 0.1393    | 0.4542 |
| ISOP | 0.0634  | 0.5104 | 0.0288    | 0.2320 |
| SQT  | 1.537   | -      | 1.537     | -      |

5    * as used in Ying et al. (2015)





**Table 2 Predicted seasonal average component contributions (%) to SOA. SEMI: semi-volatile isoprene SOA; OLGM: oligomers; IEPOX: SOA from isoprene epoxide; MAE: SOA from methacrylic acid epoxide; GLY: SOA from glyoxal; MGLY: SOA from methylglyoxal.**

|        | **Spring** | **Summer** | **Autumn** | **Winter** |
|--------|--------|--------|--------|--------|
| SEMI   | 19.8%  | 22.0%  | 22.1%  | 20.6%  |
| OLGM   | 26.7%  | 20.8%  | 22.7%  | 26.8%  |
| IEPOX  | 13.3%  | 14.3%  | 8.6%   | 4.5%   |
| MAE    | 0.7%   | 0.8%   | 0.6%   | 0.3%   |
| GLY    | 14.4%  | 14.2%  | 19.4%  | 24.5%  |
| MGLY   | 25.1%  | 28.0%  | 26.7%  | 23.3%  |



**Table 3 Predicted seasonal average precursor contributions (%) to SOA. ISOP: isoprene; TERP: monoterpenes; SESQ: sesquiterpenes; ARO1 and ARO2: aromatic compounds; ALK5: long-chain alkanes; GLY: primary glyoxal; MGLY: primary methylglyoxal; and BSOA: SOA due to biogenic emissions (ISOP+TERP+SESQ).**

|        | Spring | Summer | Autumn | Winter |
|--------|--------|--------|--------|--------|
| ISOP   | 47.8%  | 60.7%  | 36.0%  | 15.9%  |
| TERP   | 10.0%  | 10.3%  | 10.8%  | 7.4%   |
| SESQ   | 1.6%   | 3.6%   | 2.6%   | 0.5%   |
| ARO1   | 13.2%  | 6.8%   | 13.3%  | 15.2%  |
| ARO2   | 7.8%   | 7.6%   | 13.4%  | 18.7%  |
| ALK5   | 8.2%   | 3.9%   | 9.5%   | 17.5%  |
| GLY    | 6.9%   | 5.4%   | 10.4%  | 17.1%  |
| MGLY   | 4.4%   | 1.8%   | 4.1%   | 7.7%   |
| *BSOA* | *59.5%* | *74.5%* | *49.4%* | *23.9%* |





**Table 4 Settings for sensitivity simulations.**

| Simulation | Abbreviation | Notes |
|---|---|---|
| 1 | Base | Base cases simulation; all biogenic and anthropogenic emissions at 100% |
| 2 | 0.5NOx | Anthropogenic NOx reduced by 50% |
| 3 | 0.75VOC | Anthropogenic VOC reduced by 25% |
| 4 | NV | Anthropogenic NOx and VOC reduced by 50% and 25%, respectively |
| 5 | 0.5ISOP | Biogenic isoprene emission reduced by 50% |
| 6 | BIO | Biogenic emissions at 100%; no anthropogenic emissions |





Figure 1 Predicted and observed August VOC concentrations in Nanjing. Observed concentrations are based on individual VOCs lumped into the SAPRC-11 model species. Shaded areas represent the range of predictions in the 9 grid cells with the monitor site at the center. Units are ppb.



**Figure 2 Predicted and observed concentrations of total organic carbon (OC, left column) and elemental carbon (EC, right column) in several different cities. BH: Beihang University campus; TH: Tsinghua University campus; Nanjing: Jiangsu Provincial Academy of Environmental Science; Guangzhou: Tianhu (rural site). Shaded areas represent the range of predictions in the 9 grid cells with the monitor site at the center. Units are μg m-3.**



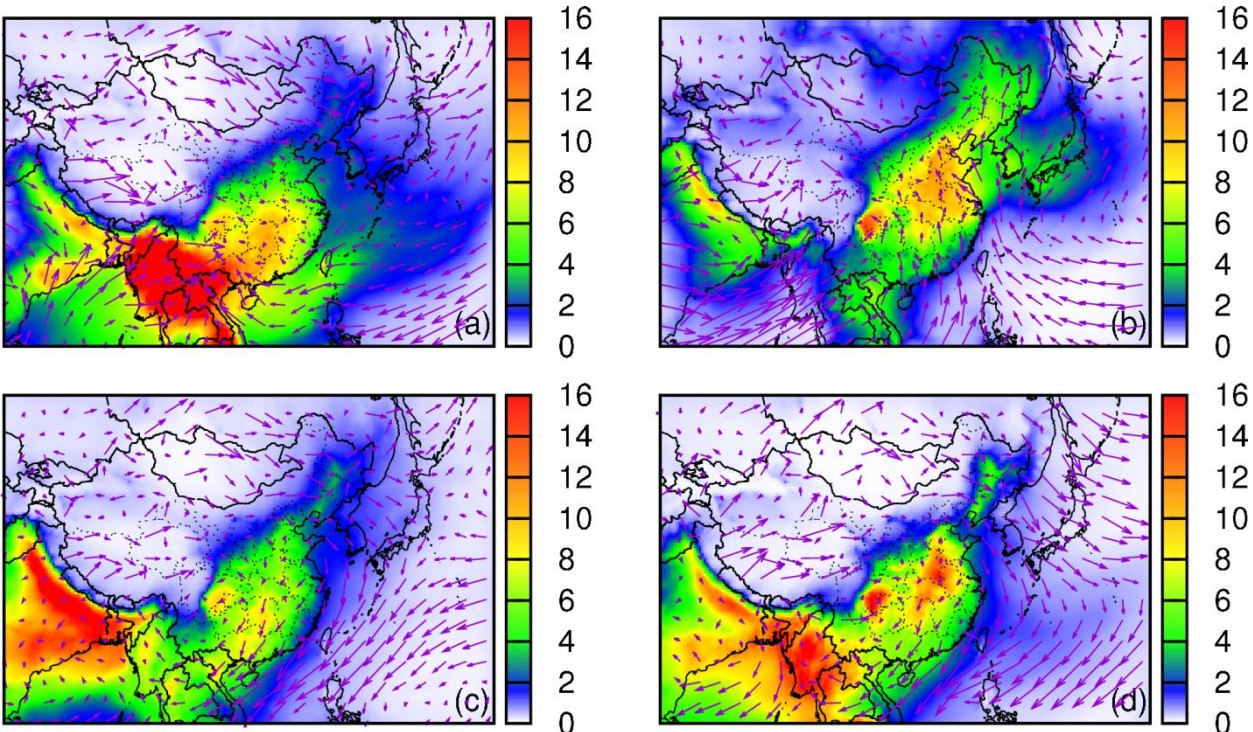

**Figure 3 Predicted monthly average SOA concentrations in (a) spring (March, April and May), (b) summer (June, July and August), (c) fall (September, October and November), and (d) winter (December, January and February) 2013. Units are µg m-3. Overlaid on the panels are seasonal average wind vectors.**





**Figure 4 Predicted seasonal average SOA component concentrations in (a) spring, (b) summer, (c) autumn, and (d) winter 2011. Units are µg m⁻³. SEMI: semi-volatile SOA; OLGM: oligomers; IEPOX: SOA from isoprene expoxide; MAE: SOA from methacrylic acid epoxide; GLY: SOA from glyoxal; MGLY: SOA from methylgloxoal.**





**Figure 5 Predicted seasonal average SOA concentrations due to contributions from different precursors in (a) spring (March, April and May), (b) summer (June, July and August), (c) autumn (September, October and November), and (d) winter (December, January and February) 2013. Units are µg m-3. ARO: aromatics; ALK: alkane; ISOP: isoprene; TERP: monoterpenes; SESQ: sesquiterpenes; [M]GLY: primary glyoxal+methylglyoxal**




**Figure 6 Predicted time series of SOA concentrations (secondary y axis, units are µg m-3), and fractional contributions to SOA due to different precursors at four cities (Beijing, Shanghai, Guangzhou and Chengdu) in January and August, 2013. Units are µg m-3.**
5     **ARO: aromatics; ISOP: isoprene; TERP: monoterpene; SESQ: sesquiterpene; [M]GLY: glyoxal+methylglyoxal directly emitted or from other precursors; OTHER: boundary conditions**



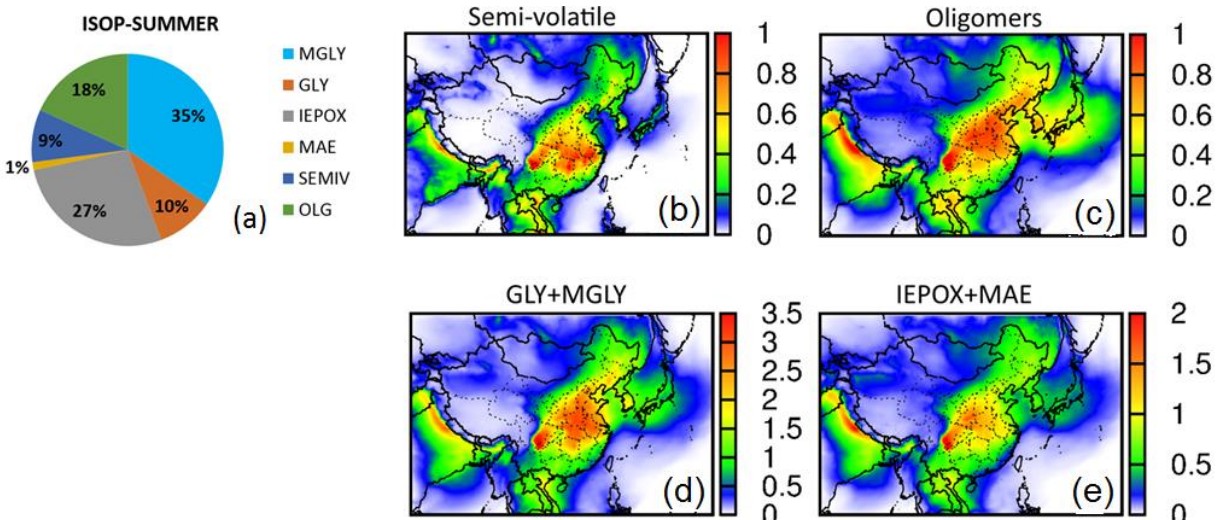

**Figure 7 Predicted fractional component contributions to summertime isoprene SOA in China (Panel a) and spatial distributions of iSOA from the traditional (b and c) and surface uptake (d and e) pathways. Units are µg m⁻³ for panels b-d.**





**Figure 8 Predicted (a) spatial distribution of SOA for August 2013, and change of SOA concentrations (sensitivity case – base case) due to (b) reduction of anthropogenic NOx by 50%, (c) reduction of anthropogenic VOC by 25%, (d) simultaneous reduction of anthropogenic NOx and VOC by 50% and 25%, (e) reduction of biogenic isoprene by 50% and (f) reduction of all anthropogenic emissions.**



**Figure 9 Same as Figure 8, for January 2013.**