# Peer review of "Modeling Biogenic and Anthropogenic Secondary Organic Aerosol in China"

_Atmospheric Chemistry and Physics, 2016_

## Referee Comment (RC1) · Anonymous Referee #1 · 18 Aug 2016

Hu et al. present a regional modeling study for China in 2013 focusing on contributions to secondary organic aerosol (SOA). They consider some more recently recognized pathways to SOA such as heterogeneous uptake of epoxides, dicarbonyls, and oligomerization in addition to traditional semivolate SOA. SOA is classified in terms of its parent hydrocarbon source as well as precursor in different seasons across the domain. Model predictions of OC as well as precursor gases are compared to observations in select locations at select times and the model seems to perform reasonably.

Main comments

1. MGLY SOA: This work predicts a large role for methylglyoxal (MGLY) in forming SOA (23-28% of SOA), consistent with their previous work for the eastern United States (Ying et al., 2015). How well is this supported by laboratory and/or field work? Is the

[Figure]

MGLY parameterization justified given that the uptake coefficient is based on glyoxal? More recent work by Marais et al. (2016) scaled the MGLY uptake coefficient to that of glyoxal using the relative Henry's law coefficient resulting in MGLY producing less than 1% of isoprene SOA. Mechanistic modeling by Woo and McNeill (2015) also indicate MGLY is not a dominant contributor to SOA.

2. Biogenic vs anthropogenic carbon and POA vs SOA: This work's predictions of SOA indicate a significant fraction of SOA contains modern carbon as it comes from biogenic VOCs such as isoprene and monoterpenes. Total OA in the study is however dominated by POA (SOA is ~30% of total OA, Fig S5). Other recent work such as that of Zhao et al. (2016) indicates anthropogenic VOCs (specifically semivolatile POA and IVOCs) are the major contributors to SOA in China. Can the authors reconcile their results with Zhao et al.'s results? Can the authors provide any insight as to why their large modern carbon contribution is more (or less) accurate than the anthropogenic VOC hypothesis? This affects your control strategy and which VOCs you might target (ie those important for OH interactions or those with low-volatility). Are there modern/fossil carbon measurements or POA/SOA proxies that can be compared with the model?

3. While the figures are clear and nicely presented, there could be more synthesis of information in the figures. Figure 2 for example has different dates in each panel and different vertical axis limits as well. The last figure shows some synthesis by including a pie chart along with spatial distribution. Figure S2 (locations) would be best in the main manuscript. Figure 4-5 each have 24 subplots. While the information is useful and I don't recommend removing it, it would be nice to have synthesis plots too. As an example, do underestimates in any of the precursor species correlate with underestimates in OC?

Other comments

4. Recent work by Marais et al. (2016) and Lin et al. (2016) indicate IEPOX SOA

is mainly controlled via aerosol surface area which is linked to sulfate. The author's mechanism of IEPOX uptake may capture this phenomenon and show a relationship with sulfate. Page 15, line 5 about the model not capturing the Xu et al. relationship with sulfate should be verified.

5. Page 1, line 26 indicates SOA is highest in summer, but this seems very spatially dependent with winter perhaps having higher concentrations in a more localized area. Clarify.

6. Page 4, Model description section: Are these simulations the same as used by Hu et al. 2016?

7. Page 5, line 2: What CMAQ version served as the basis for this work?

8. Page 6, 7 and for data in general, can you provide a latitude, longitude, and sampling altitude for observations? Will observational data be made available with this manuscript for future model evaluation?

9. Page 6, line 13: Is there a reference for the PAMS method?

10. Page 6, line 28: Which species in particular are you referring to in terms of good olefin performance? OLE2 was quite high.

11. Page 7, line 7: In light of potentially large vehicle contributions to isoprene mentioned here, in your work, is isoprene attributed entirely (or mostly) to biogenic sources?

12. Page: 8, line 25, regarding underestimated OC, what about potential missing SOA sources (such as IVOCs, etc)? What role may they play? See also main comment number 2.

13. Page 19, Table 1: How do ARO1 and ARO2 map to benzene, toluene, and xylene and their respective yields (as used in CMAQ v4.7 and later)? C* should be provided with the alphas.

14. Page 19, Table 1: This table indicates the aromatic alphas were increased 13%

while the monoterpene alphas were increased 30%. Isoprene alphas were increased by 2.2x. These numbers are all consistent with the biases in high-NOx SOA yields reported by Zhang et al. (2014). As Zhang et al. reported the bias in yield, it is the yield not alpha that should be increased which involves refitting the yield vs organic aerosol concentration data to get the new alpha and C* parameters. Scaling the alpha alone results in an upper bound correction. The wall loss corrections have also been shown to be highly chamber specific (for example, Zhang et al. report two different toluene yield factors: 2.2 (their work) and 1.13 (another study)). Are the original parameterizations and the correction values from Zhang et al. from the same group/chamber? TERP yields in the original formulation match Carlton et al. (2010) and thus were a weighted contribution from different monoterpenes in the work of Griffin et al. (1999). Zhang et al. a-pinene+OH matches work from Chhabra et al. (2011). I suspect performing the proper correction to yield curves is unlikely to significantly change conclusions, but we should avoid propagating incorrect values.

15. Page 22, Table 4, Simulation 6: Clarify that anthropogenic VOC, NOx, SO2, etc were removed (not just VOC, NOX)

16. What is the major driver for how anthropogenic emissions affect SOA? Is it through POA?

References

Carlton, A. G., Bhave, P. V., Napelenok, S. L., Edney, E. D., Sarwar, G., Pinder, R. W., Pouliot, G. A., and Houyoux, M.: Model representation of secondary organic aerosol in CMAQv4.7, Environ. Sci. Technol., 44, 8553-8560, 10.1021/es100636q, 2010.

Griffin, R. J., Cocker, D. R., Flagan, R. C., and Seinfeld, J. H.: Organic aerosol formation from the oxidation of biogenic hydrocarbons, J. Geophys. Res., 104, 3555-3567, 1999.

Lin, G., Penner, J. E., and Zhou, C.: How will SOA change in the future?, Geophys.

Res. Lett., 43, 1718-1726, 10.1002/2015GL067137, 2016.

Marais, E. A., Jacob, D. J., Jimenez, J. L., Campuzano-Jost, P., Day, D. A., Hu, W., Krechmer, J., Zhu, L., Kim, P. S., Miller, C. C., Fisher, J. A., Travis, K., Yu, K., Hanisco, T. F., Wolfe, G. M., Arkinson, H. L., Pye, H. O. T., Froyd, K. D., Liao, J., and McNeill, V. F.: Aqueous-phase mechanism for secondary organic aerosol formation from isoprene: application to the southeast United States and co-benefit of SO2 emission controls, Atmos. Chem. Phys., 16, 1603-1618, 10.5194/acp-16-1603-2016, 2016.

Woo, J. L., and McNeill, V. F.: simpleGAMMA v1.0 – a reduced model of secondary organic aerosol formation in the aqueous aerosol phase (aaSOA), Geosci. Model Dev., 8, 1821-1829, 10.5194/gmd-8-1821-2015, 2015.

Ying, Q., Li, J., and Kota, S. H.: Significant contributions of isoprene to summertime secondary organic aerosol in eastern United States, Environ. Sci. Technol., 49, 7834-7842, 10.1021/acs.est.5b02514, 2015.

Zhang, X., Cappa, C. D., Jathar, S. H., McVay, R. C., Ensberg, J. J., Kleeman, M. J., and Seinfeld, J. H.: Influence of vapor wall loss in laboratory chambers on yields of secondary organic aerosol, Proc. Natl. Acad. Sci. U. S. A., 111, 5802-5807, 10.1073/pnas.1404727111, 2014.

Zhao, B., Wang, S., Donahue, N. M., Jathar, S. H., Huang, X., Wu, W., Hao, J., and Robinson, A. L.: Quantifying the effect of organic aerosol aging and intermediate-volatility emissions on regional-scale aerosol pollution in China, Sci. Rep., 6, 28815, 10.1038/srep28815, 2016.
* * *

---

## Referee Comment (RC2) · Anonymous Referee #2 · 3 Oct 2016

This manuscript presents in detail the predicted secondary organic aerosol in China by a regional CTM model. Model performance is evaluated by comparing the predicted VOC, EC, and OC concentrations to the observations at several urban sites. In general, the paper is well-written. However, it is well known that SOA is complex and the parameterization of various pathways is highly uncertain. There are limited discussion and sensitivity tests on the uncertainty of the presented pathways as well as the potential contribution of unaccounted pathways. The results of the paper are also lack of observational constraints. I suggest the following comments to be considered for revision.

Specific comments:

(1) Uncertainty of SOA pathways:

[Figure]

– GLY/MGLY uptake is found to a major contributor to SOA in this study. The simulation is however based on irreversible uptake, whilst some studies have shown the process may also be reversible (Fu et al., 2009). Large uncertainties remain in the uptake mechanism (Galloway et al., 2011). A short summary of laboratory findings is needed along with the discussion about the potential impact on the model predictions.

– Isoprene seems extremely important given the predictions (contributing 61% in the summertime). The predicted isoprene however doesn't match well with the observations (seems being over-predicted). The fact that better agreements were found for MACR and MVK makes the case more complicated (i.e., possibly underestimation of OH). I think it is important to confirm that the predicted isoprene peak is not due to errors in the model. Either comparison to other locations with the same model or other model predictions using the same version of MEGAN is needed.

– The SOA predictions are lack of observational constraints. Intensive work has been done in major cities in China for example by AMS, which provides details about various OA types and estimates of the oxidation state (Hu et al., 2016; Huang et al., 2013; Sun et al., 2016). There are also off-line filter measurements, e.g., Yang et al. (2016) and He et al. (2014) for SOA tracers in China. Direct comparisons to such observations may be difficult to make, but rough comparisons may provide constraints to tell if the model predictions make sense, and if different studies are consistent. For example, the oxidation state of the OA as well as the organosulfate concentration may not support a high contribution of oligomers as well as GLY/MGLY/IEPOX SOA.

– How could the unaccounted processes change the predictions? For example, the conversion between POA to SOA (Robinson et al., 2007), the gaseous (Donahue et al., 2012) and heterogeneous aging of SOA (Kroll et al., 2015).

(2) Page 4, Line 30-32: The isoprene SOA yield does not vary much for dry or wet conditions (Carlton et al., 2009) as well as solid or liquid seed particles (Kuwata et al., 2012). Acidity is a key factor that regulate the production of SOA from isoprene (Kuwata

et al., 2015). In fact, the yields from Kroll et al. (2006) along with other chamber studies represent both contributions of traditional partitioning and reactive uptake, although the contribution of reactive uptake might be much smaller compared to acidic conditions.

(3) Page 5, Line 7: Which version of MEIC inventory (or the base year) has been used? Isn't MEIC only for the mainland China but not for the surrounding areas in East and Southeast Asia?

(4) Descriptions about the observations are needed in Section 2, particularly for unpublished data. In Sect. 3.1.1., does the VOC data come from GC-MS measurements? What is the time resolution? Have the data been processed? For the EC/OC data, what is the time resolution? Are the data from different studies obtained and processed in the same way?

(5) Page 6, Line 24: What kind of measurement errors?

(6) Page 8, Line 13-15: This is somewhat misleading. EC can't represent all primary emissions.

(7) Page 8, Line 30-35: It is well known that POC themselves can be semi-volatile and form SOA quickly (Robinson et al., 2007). I think the data are too limited to achieve a conclusion that POC or OC is well estimated by the model.

(8) Page 9, Line 6-7: Please clarify whether the open biomass burning emissions contribute directly more SOA precursors or more POC that causes more gas-to-particle condensation.

(9) Page 10, Line 10-15: It is also because the aging of SOA is not considered in the model.

Technical Remarks:

Page 10, Line 10: "sptial" should be "spatial".

Reference:

[Figure]

Carlton, A. G., Wiedinmyer, C., and Kroll, J. H.: A review of secondary organic aerosol (SOA) formation from isoprene, Atmos. Chem. Phys., 9, 4987-5005, 2009.

Donahue, N. M., Henry, K. M., Mentel, T. F., Kiendler-Scharr, A., Spindler, C., Bohn, B., Brauers, T., Dorn, H. P., Fuchs, H., Tillmann, R., Wahner, A., Saathoff, H., Naumann, K. H., Mohler, O., Leisner, T., Muller, L., Reinnig, M. C., Hoffmann, T., Salo, K., Hallquist, M., Frosch, M., Bilde, M., Tritscher, T., Barmet, P., Praplan, A. P., DeCarlo, P. F., Dommen, J., Prevot, A. S. H., and Baltensperger, U.: Aging of biogenic secondary organic aerosol via gas-phase OH radical reactions, Proc. Natl. Acad. Sci. U. S. A., 109, 13503-13508, 10.1073/pnas.1115186109, 2012.

Fu, T. M., Jacob, D. J., and Heald, C. L.: Aqueous-phase reactive uptake of dicarbonyls as a source of organic aerosol over eastern North America, Atmos. Environ., 43, 1814-1822, 10.1016/j.atmosenv.2008.12.029, 2009.

Galloway, M. M., Loza, C. L., Chhabra, P. S., Chan, A. W. H., Yee, L. D., Seinfeld, J. H., and Keutsch, F. N.: Analysis of photochemical and dark glyoxal uptake: Implications for SOA formation, Geophys. Res. Lett., 38, L17811, 10.1029/2011gl048514, 2011.

He, Q. F., Ding, X., Wang, X. M., Yu, J. Z., Fu, X. X., Liu, T. Y., Zhang, Z., Xue, J., Chen, D. H., Zhong, L. J., and Donahue, N. M.: Organosulfates from pinene and isoprene over the Pearl River Delta, South China: Seasonal variation and implication in formation mechanisms, Environ. Sci. Technol., 48, 9236-9245, 10.1021/es501299v, 2014.

Hu, W. W., Hu, M., Hu, W., Jimenez, J. L., Yuan, B., Chen, W. T., Wang, M., Wu, Y. S., Chen, C., Wang, Z. B., Peng, J. F., Zeng, L. M., and Shao, M.: Chemical composition, sources, and aging process of submicron aerosols in Beijing: Contrast between summer and winter, J. Geophys. Res., 121, 1955-1977, 10.1002/2015jd024020, 2016.

Huang, X. F., Xue, L., Tian, X. D., Shao, W. W., Sun, T. L., Gong, Z. H., Ju, W. W., Jiang, B., Hu, M., and He, L. Y.: Highly time-resolved carbonaceous aerosol characterization

in Yangtze River Delta of China: Composition, mixing state and secondary formation, Atmos. Environ., 64, 200-207, 10.1016/j.atmosenv.2012.09.059, 2013.

Kroll, J. H., Lim, C. Y., Kessler, S. H., and Wilson, K. R.: Heterogeneous oxidation of atmospheric organic aerosol: Kinetics of changes to the amount and oxidation state of particle-phase organic carbon, J. Phys. Chem. A, 119, 10767-10783, 10.1021/acs.jpca.5b06946, 2015.

Kuwata, M., Zorn, S. R., and Martin, S. T.: Using elemental ratios to predict the density of organic material composed of carbon, hydrogen, and oxygen, Environ. Sci. Technol., 46, 787-794, 10.1021/es202525q, 2012.

Kuwata, M., Liu, Y., McKinney, K. A., and Martin, S. T.: Physical state and acidity of inorganic sulfate can regulate the production of secondary organic material from isoprene photooxidation products, Phys. Chem. Chem. Phys., 17, 5670-5678, 10.1039/C4CP04942J, 2015.

Robinson, A. L., Donahue, N. M., Shrivastava, M. K., Weitkamp, E. A., Sage, A. M., Grieshop, A. P., Lane, T. E., Pierce, J. R., and Pandis, S. N.: Rethinking organic aerosols: Semivolatile emissions and photochemical aging, Science, 315, 1259-1262, 10.1126/science.1133061, 2007.

Sun, Y. L., Du, W., Fu, P. Q., Wang, Q. Q., Li, J., Ge, X. L., Zhang, Q., Zhu, C. M., Ren, L. J., Xu, W. Q., Zhao, J., Han, T. T., Worsnop, D. R., and Wang, Z. F.: Primary and secondary aerosols in Beijing in winter: sources, variations and processes, Atmos. Chem. Phys., 16, 8309-8329, 10.5194/acp-16-8309-2016, 2016.

Yang, F., Kawamura, K., Chen, J., Ho, K. F., Lee, S. C., Gao, Y., Cui, L., Wang, T. G., and Fu, P. Q.: Anthropogenic and biogenic organic compounds in summertime fine aerosols (PM2.5) in Beijing, China, Atmos. Environ., 124, 166-175, 10.1016/j.atmosenv.2015.08.095, 2016.

---

## Author Comment (AC1) · 15 Nov 2016

This manuscript presents in detail the predicted secondary organic aerosol in China by a regional CTM model. Model performance is evaluated by comparing the predicted VOC, EC, and OC concentrations to the observations at several urban sites. In general, the paper is well-written. However, it is well known that SOA is complex and the parameterization of various pathways is highly uncertain. There are limited discussion and sensitivity tests on the uncertainty of the presented pathways as well as the potential contribution of unaccounted pathways. The results of the paper are also lack of observational constraints. I suggest the following comments to be considered for revision.

Response: Thank you for the comments to help improve the quality of the paper. We

have revised the manuscript to address your comments and a detailed response to each comment is provided in this file.

Specific comments: (1) Uncertainty of SOA pathways: – GLY/MGLY uptake is found to a major contributor to SOA in this study. The simulation is however based on irreversible uptake, whilst some studies have shown the process may also be reversible (Fu et al., 2009). Large uncertainties remain in the uptake mechanism (Galloway et al., 2011). A short summary of laboratory findings is needed along with the discussion about the potential impact on the model predictions.

Responses: A few sentences were included on page 5, lines 1-3 to discuss this: "The treatment of GLY and MGLY SOA formation as an irreversible process provides an upper-limit estimation of SOA formation from these two precursors, as it has been reported that reactive uptake of GLY and MGLY can be reversible (Galloway et al., 2009) and dissolved GLY and MGLY can react with oxidants to form products with volatile products (Lim et al., 2010)."

– Isoprene seems extremely important given the predictions (contributing 61% in the summertime). The predicted isoprene however doesn't match well with the observations (seems being over-predicted). The fact that better agreements were found for MACR and MVK makes the case more complicated (i.e., possibly underestimation of OH). I think it is important to confirm that the predicted isoprene peak is not due to errors in the model. Either comparison to other locations with the same model or other model predictions using the same version of MEGAN is needed.

Responses: We have examined our model results carefully and believe that the isoprene peak is not due to errors in the model (i.e. emissions and meteorological conditions are normal). As we explained in the manuscript, due to the short lifetime of isoprene ($\sim$1-2 h) but longer lifetime of MACR and MVK (5-10 h), over-prediction of isoprene at one urban location does not necessarily mean MACR and MVK in the same grid cell must be overpredicted as well – as most of the MACR and MVK are

likely from neighboring grid cells. It would be ideal if high resolution isoprene measurements were available at another non-urban location. No changes were made regarding this comment.

– The SOA predictions are lack of observational constraints. Intensive work has been done in major cities in China for example by AMS, which provides details about various OA types and estimates of the oxidation state (Hu et al., 2016; Huang et al., 2013; Sun et al., 2016). There are also off-line filter measurements, e.g., Yang et al. (2016) and He et al. (2014) for SOA tracers in China. Direct comparisons to such observations may be difficult to make, but rough comparisons may provide constraints to tell if the model predictions make sense, and if different studies are consistent. For example, the oxidation state of the OA as well as the organosulfate concentration may not support a high contribution of oligomers as well as GLY/MGLY/IEPOX SOA.

Responses: As the reviewer pointed out, a detailed comparison of these measurements is not possible because most of them were not at made at 2013, and PM1 was measured and OOA was used as a surrogate of SOA in the AMS studies. We compared our results with the major findings in these studies and found the SOA/OA fractions in summer in our studies (50-70% in urban areas) is consistent with Huang et al (2013) ($\sim$68.3%) and Hu et al. (2016) (45-67%). Also, consistent findings were found that POA is more dominant in OA in winter in our study and in three AMS studies (Hu et al., 2016; Huang et al., 2013; Sun et al., 2016), although our estimation for the SOA/OA fractions in winter is lower than theirs. We included above comparisons in Section 3.2.1 in the revised manuscript and cited the references.

While we strongly agree with the reviewer that these measurements can be applied to provide constraint on model predictions, this is better done by comparing with local scale modeling studies with higher spatial resolutions and refined local emission inventories. More effort is needed to use them to 'constraint' the model – as no parameters in the current model and input data can be directly adjusted based on these spatially sparse measurements. In the revised manuscript, we pointed out the existence of such

**[ACPD](ACPD)**
data on page 16:

"It should be noted that the reported SOA concentrations in this study have not been compared with direct measurements of SOA. More detailed measurements of organic components within the aerosol phase, such as the oxidation state measurements from aerosol mass spectrometers (AMS) (Hu et al., 2016) and tracer species representing SOA formation from precursor species (Yang et al., 2016), are becoming available in many areas in China. Future local scale SOA modeling studies can be conducted to better utilize these data as constraints on model parameters and input data."

– How could the unaccounted processes change the predictions? For example, the conversion between POA to SOA (Robinson et al., 2007), the gaseous (Donahue et al., 2012) and heterogeneous aging of SOA (Kroll et al., 2015).

Responses: Additional studies are obviously needed to answer how these processes will change the SOA predictions. As the reviewer #1 pointed out and we have included in the revised manuscript, Zhao et al. (2016) included IVOCs, aging of POA and SOA, and found dramatic increase of OA concentrations due to these processes. In this study, we focus on the important contributions of glyoxal, methylglyoxal and isoprene epoxydiols to SOA formation, which have also been demonstrated in many studies. We are not sure how Zhao et al. handled these processes. Nonetheless, in both studies the model-measurement discrepancies cannot be fully solved, which could indicate that there might be other precursors/processes missing and merit more studies in this area. We acknowledge the potential contributions from the processes that the reviewer pointed out, and we included a short discussion in Section 3.1 in the revised manuscript.

(2) Page 4, Line 30-32: The isoprene SOA yield does not vary much for dry or wet conditions (Carlton et al., 2009) as well as solid or liquid seed particles (Kuwata et al., 2012). Acidity is a key factor that regulate the production of SOA from isoprene (Kuwata et al., 2015). In fact, the yields from Kroll et al. (2006) along with other chamber studies

represent both contributions of traditional partitioning and reactive uptake, although the contribution of reactive uptake might be much smaller compared to acidic conditions.

Responses: Points taken but no changes were necessary regarding this comment.

(3) Page 5, Line 7: Which version of MEIC inventory (or the base year) has been used? Isn't MEIC only for the mainland China but not for the surrounding areas in East and Southeast Asia?

Responses: We used MEIC inventory V1.0 in our study for China emissions and used Regional Emission inventory in ASia version 2 (REAS2) (Kurokawa et al., 2013) for other countries and regions rather than China in the domain. This is clarified on page 5, lines 16-17.

(4) Descriptions about the observations are needed in Section 2, particularly for unpublished data. In Sect. 3.1.1., does the VOC data come from GC-MS measurements? What is the time resolution? Have the data been processed? For the EC/OC data, what is the time resolution? Are the data from different studies obtained and processed in the same way?

Responses: The VOC data were measured by GC-MS at an hourly time resolution. As stated in the manuscript, the measured detailed 54 VOC species were further lumped into the SAPRC11 (Carter and Heo, 2012) species for model comparison. The site information was also included and a reference was cited for the PAMS method in the revised manuscript (see the responses to reviewer #1's comment # 8 and #9). We provided more description of the VOC data in Section 2. Details of the EC/OC measurements are provided in the cited reference. All EC/OC measurements were daily data analyzed using thermal-optical EC/OC analyzer.

(5) Page 6, Line 24: What kind of measurement errors? Responses: We don't have any evidence, so we are not sure whether it is a measurement error or what kind of measurement error could lead to increase in many VOC species. This statement is

removed in the revised manuscript to avoid conjecture with no evidence.

(6) Page 8, Line 13-15: This is somewhat misleading. EC can't represent all primary emissions.

Responses: While EC cannot represent all primary emissions, it does correlate with primary OC for most combustion sources quite well. The manuscript clearly states "primary OC' instead of 'all primary emissions'. In fact, the primary OC/EC ratio method is a well-known and widely applied technique to estimate POA and SOA split from total OC. No changes were made regarding this comment.

(7) Page 8, Line 30-35: It is well known that POC themselves can be semi-volatile and form SOA quickly (Robinson et al., 2007). I think the data are too limited to achieve a conclusion that POC or OC is well estimated by the model. Responses: The agreement of predicted with measured OC and EC at multiple sites in a few sampling periods in different seasons build confidence that model generally reproduced the concentration levels and daily variations of OC and EC, although OC or EC was underpredicted in a few high pollution episodes. The OC agreement does not directly prove that POC or SOC were predicted correctly (it could be a compensating between POC and SOC). The process that semi-volatile POC forms SOA would change the split of OC between POC and SOC, but likely would not change much of the 'carbon' concentrations. No changes were made regarding this comment.

(8) Page 9, Line 6-7: Please clarify whether the open biomass burning emissions contribute directly more SOA precursors or more POC that causes more gas-to-particle condensation.

Responses: Our analysis shows that large contributions from GLY, MGLY and IEPOX. Thus, this is directly related with large emissions of SOA precursors. This is clarified in the revised manuscript on page 9, lines 22-23.

(9) Page 10, Line 10-15: It is also because the aging of SOA is not considered in the

model.

Responses: Thanks for pointing out this possible reason. We have added it in the revised manuscript.

Technical Remarks: Page 10, Line 10: "sptial" should be "spatial".

Responses: This error has been corrected.

References: Carter, W. P. L., and Heo, G.: Development of revised SAPRC aromatics mechanisms. Final Report to the California Air Resources Board, Contracts No. 07-730 and 08-326, April 12, 2012. , 2012. Galloway, M. M., Chhabra, P. S., Chan, A. W. H., Surratt, J. D., Flagan, R. C., Seinfeld, J. H., and Keutsch, F. N.: Glyoxal uptake on ammonium sulphate seed aerosol: reaction products and reversibility of uptake under dark and irradiated conditions, Atmos. Chem. Phys., 9, 3331-3345, 10.5194/acp-9-3331-2009, 2009. Hu, W., Hu, M., Hu, W., Jimenez, J. L., Yuan, B., Chen, W., Wang, M., Wu, Y., Chen, C., Wang, Z., Peng, J., Zeng, L., and Shao, M.: Chemical composition, sources, and aging process of submicron aerosols in Beijing: Contrast between summer and winter, Journal of Geophysical Research: Atmospheres, 121, 1955-1977, 10.1002/2015jd024020, 2016. Kurokawa, J., Ohara, T., Morikawa, T., Hanayama, S., Janssens-Maenhout, G., Fukui, T., Kawashima, K., and Akimoto, H.: Emissions of air pollutants and greenhouse gases over Asian regions during 2000–2008: Regional Emission inventory in ASia (REAS) version 2, Atmos. Chem. Phys., 13, 11019-11058, 10.5194/acp-13-11019-2013, 2013. Lim, Y. B., Tan, Y., Perri, M. J., Seitzinger, S. P., and Turpin, B. J.: Aqueous chemistry and its role in secondary organic aerosol (SOA) formation, Atmos. Chem. Phys., 10, 10521-10539, 10.5194/acp-10-10521-2010, 2010. Yang, F., Kawamura, K., Chen, J., Ho, K., Lee, S., Gao, Y., Cui, L., Wang, T., and Fu, P.: Anthropogenic and biogenic organic compounds in summertime fine aerosols (PM2.5) in Beijing, China, Atmospheric Environment, 124, Part B, 166-175, http://dx.doi.org/10.1016/j.atmosenv.2015.08.095, 2016.

---

## Author Comment (AC3) · 15 Nov 2016

Response to Anonymous Referee #1

Hu et al. present a regional modeling study for China in 2013 focusing on contributions to secondary organic aerosol (SOA). They consider some more recently recognized pathways to SOA such as heterogeneous uptake of epoxides, dicarbonyls, and oligomerization in addition to traditional semi-volatile SOA. SOA is classified in terms of its parent hydrocarbon source as well as precursor in different seasons across the domain. Model predictions of OC as well as precursor gases are compared to observations in select locations at select times and the model seems to perform reasonably.

Main comments 1. MGLY SOA: This work predicts a large role for methylglyoxal (MGLY) in forming SOA (23-28% of SOA), consistent with their previous work for the

eastern United States (Ying et al., 2015). How well is this supported by laboratory and/or field work? Is the MGLY parameterization justified given that the uptake coefficient is based on glyoxal? More recent work by Marais et al. (2016) scaled the MGLY uptake coefficient to that of glyoxal using the relative Henry's law coefficient resulting in MGLY producing less than 1% of isoprene SOA. Mechanistic modeling by Woo and McNeill (2015) also indicate MGLY is not a dominant contributor to SOA.

Responses: Measurement of uptake coefficient of methylglyoxal on acidic particles has been reported by Zhao et al. (2006). While the effective Henry's law coefficient determined from that experiment is on the same order of magnitude as cited in Marais et al. (2016), the measured uptake coefficients for methylglyoxal was on the order of 7.6×10-3, which is on the same order of magnitude as those of glyoxal on acidic particles. Assuming that the uptake coefficient can be scaled by effective Henry's law alone might be an over-simplification and has not verified experimentally. In the revised manuscript in Section 4.1, we mentioned the work of Marais et al. (2016) as well as Zhao et al. (2006) to inform the readers that significant uncertainties can exist in the uptake coefficient.

2. Biogenic vs anthropogenic carbon and POA vs SOA: This work's predictions of SOA indicate a significant fraction of SOA contains modern carbon as it comes from biogenic VOCs such as isoprene and monoterpenes. Total OA in the study is however dominated by POA (SOA is _30% of total OA, Fig S5). Other recent work such as that of Zhao et al. (2016) indicates anthropogenic VOCs (specifically semi-volatile POA and IVOCs) are the major contributors to SOA in China. Can the authors reconcile their results with Zhao et al.'s results? Can the authors provide any insight as to why their large modern carbon contribution is more (or less) accurate than the anthropogenic VOC hypothesis? This affects your control strategy and which VOCs you might target (i.e. those important for OH interactions or those with low-volatility). Are there modern/fossil carbon measurements or POA/SOA proxies that can be compared with the model?

Responses: The significance of modern carbon to SOA varies significantly from season to season. As indicated in Table 3, contributions of BSOA to total SOA varies from 24% (in winter) to 75% (in summer) based on average concentrations throughout the country. Zhao et al. (2016), however, estimated a much lower contribution of BSOA to total SOA.

Although Zhao et al. (2016) showed that the "high-yield" VBS model led to higher SOA predictions, it is unclear to the authors how the model handles the important contributions of glyoxal, methylglyoxal and isoprene epoxydiols to SOA formation. The typical reactive uptake approach does not fit the VBS modeling framework. The enhanced yields used in the VBS approach for anthropogenic emissions and IVOCs to improve the model prediction and observation of OA might lead to an overestimation of the importance of the anthropogenic contributions to SOA. A recent study reveals that in addition to primary emissions from coal burning, traffic- related exhaust and biomass burning, secondary PM (i.e. secondary organic matter and secondary inorganic matter) is an important if not a dominant contributor in 4 Chinese megacities of Beijing, Xi'an, Shanghai and Guangzhou during winter 2013 (Zhang et al., 2015). Non-fossil source contributes to ~60% of SOA or ~50% of OA. This obviously contradicts the conclusions of Zhao et al. (2006) that fossil-fuel sources always dominate the SOA budget throughout the country.

We included a short discussion on the potential contributions of the precursors/pathways in Section 3.1 in the revised manuscript on page 9.

3. While the figures are clear and nicely presented, there could be more synthesis of information in the figures. Figure 2 for example has different dates in each panel and different vertical axis limits as well. The last figure shows some synthesis by including a pie chart along with spatial distribution. Figure S2 (locations) would be best in the main manuscript. Figure 4-5 each have 24 subplots. While the information is useful and I don't recommend removing it, it would be nice to have synthesis plots too. As an example, do underestimates in any of the precursor species correlate with

underestimates in OC?

Responses: 1) Different axis limits were used to better illustrate variations in the concentrations – no changes were made regarding this point. 2) We moved Figure S2 to the main manuscript as Figure 1 3) We do not think multiple panel plot is a problem. It is the best way to illustrate the spatial distribution of different precursors and chemical components to SOA at different seasons. – no changes were made regarding this comment. 4) Unfortunately, we don't have simultaneous measurement of VOC precursors and OC at any locations.

Other comments

4. Recent work by Marais et al. (2016) and Lin et al. (2016) indicate IEPOX SOA is mainly controlled via aerosol surface area which is linked to sulfate. The author's mechanism of IEPOX uptake may capture this phenomenon and show a relationship with sulfate. Page 15, line 5 about the model not capturing the Xu et al. relationship with sulfate should be verified.

Responses: The reviewer might have misread the statement of page 15, line 5. That sentence simply pointed out that the model does not explicitly consider the effect of sulfate and nitrate on iSOA (although the acidity dependent uptake coefficient is partially related with sulfate concentrations). We change the word "modeled" to "included" in the revised manuscript.

5. Page 1, line 26 indicates SOA is highest in summer, but this seems very spatially dependent with winter perhaps having higher concentrations in a more localized area. Clarify.

Responses: We clearly mentioned that it is "generally" higher in the original statement. In the revised manuscript, the sentence in question was modified to include ", although the relative importance varies in different regions" to make it more clear.

6. Page 4, Model description section: Are these simulations the same as used by Hu

et al. 2016?

Responses: Yes, the simulations are the same as used by Hu et al. 2016. – no changes were made regarding this comment.

7. Page 5, line 2: What CMAQ version served as the basis for this work?

Responses: The CMAQ version is 5.0.1. We have added the version information in the manuscript.

8. Page 6, 7 and for data in general, can you provide a latitude, longitude, and sampling altitude for observations? Will observational data be made available with this manuscript for future model evaluation?

Responses: The latitude and longitude of the site are 33.205 N and 118.727 E, and the sampling altitude is about 15m above the ground. The observational data will be available with the manuscript upon written request to the corresponding author.

9. Page 6, line 13: Is there a reference for the PAMS method?

Responses: A reference (Lonneman, 1994) was added for the PAMS method. Detailed description of the PAMS program, including sampling methods can be found from the US EPA's website https://www3.epa.gov/ttnamti1/pamsmain.html.

10. Page 6, line 28: Which species in particular are you referring to in terms of good olefin performance? OLE2 was quite high.

Responses: The model predicted higher OLE2 on Aug. 25 and Aug. 29-30, but for other days, the model predictions were in good agreement with observations. The model predicted higher OLE1 on Aug. 2-3, Aug. 8-12, and Aug.29-30, predicted lower OLE1 on Aug. 19-21, and model predictions were in general agreement on other days. We modified "good agreement" to "general agreement" in the revised manuscript.

11. Page 7, line 7: In light of potentially large vehicle contributions to isoprene mentioned here, in your work, is isoprene attributed entirely (or mostly) to biogenic sources?

Responses: We checked the emissions in our study. Over the entire China most iso-prene emission (> 99%) is from the biogenic sources, but in urban areas, there is some isoprene emission from vehicle sources. The BSOA we mentioned in the paper includes all isoprene SOA, as we didn't track the emissions of isoprene from vehicle sources separately from biogenic sources. This could lead to some overestimation of BSOA contributions, especially in the urban areas. A future study with carbon source tracking will determine the contributions of different sources of isoprene to iSOA. No changes were made in the manuscript regarding this comment.

12. Page: 8, line 25, regarding underestimated OC, what about potential missing SOA sources (such as IVOCs, etc)? What role may they play? See also main comment number 2.

Responses: The model underpredicted both EC and OC in Beijing during the first week of March. Missing IVOC could be a potential source of underestimation of SOA in our study, but we think some primary emission sources (which emitted both EC, POA and VOC/SVOCs/IVOCs) were missing during this period. We have added the discussion on the implications of IVOCs. Also, as we discussed in the manuscript, "POA provides the medium for the partitioning of SVOCs formed from oxidation of the precursors, and correctly predicting POA is necessary for correctly predicting SOA con-centrations." Therefore, missing POA sources could also contribute to underprediction of SOA. Missing other SOA precursors in the study such as PAHs (Zhang and Ying, 2012) could also contribute to the OC under-prediction. Both field observations and de-tailed modeling studies are needed to close the gap between predicted and observed SOA concentrations. This discussion is included in the revised manuscript on page 9, lines 11-14.

13. Page 19, Table 1: How do ARO1 and ARO2 map to benzene, toluene, and xylene and their respective yields (as used in CMAQ v4.7 and later)? C* should be provided with the alphas.

[Figure]

Responses: In SAPRC11 mechanism, Aromatics with kOH < 2x104 ppm-1 min-1 are lumped as ARO1, and Aromatics with kOH > 2x104 ppm-1 min-1 are lumped as ARO2. Their yields were described in CMAQ5.0.1. C* values are provided in the table in the revised manuscript. Emissions of benzene were lumped into ARO1.

14. Page 19, Table 1: This table indicates the aromatic alphas were increased 13% while the monoterpene alphas were increased 30%. Isoprene alphas were increased by 2.2x. These numbers are all consistent with the biases in high-NOx SOA yields reported by Zhang et al. (2014). As Zhang et al. reported the bias in yield, it is the yield not alpha that should be increased which involves refitting the yield vs organic aerosol concentration data to get the new alpha and C* parameters. Scaling the alpha alone results in an upper bound correction. The wall loss corrections have also been shown to be highly chamber specific (for example, Zhang et al. report two different toluene yield factors: 2.2 (their work) and 1.13 (another study)). Are the original parameterizations and the correction values from Zhang et al. from the same group/chamber? TERP yields in the original formulation match Carlton et al. (2010) and thus were a weighted contribution from different monoterpenes in the work of Griffin et al. (1999). Zhang et al. a-pinene+OH matches work from Chhabra et al. (2011). I suspect performing the proper correction to yield curves is unlikely to significantly change conclusions, but we should avoid propagating incorrect values.

Responses: We completely agree with the reviewer that C* values should be recalculated along with the updates on the alpha values. In the revised manuscript, as commented by the reviewer, we noted that keeping the original C* values likely provide an upper bound of the wall loss effect. The mass yields and C* values used in CMAQ for toluene and xylene were based on the chamber experiments reported by Ng et al. (2007). The correction factor (1.13) provided in Table 1 of Zhang et al. (2014) are also based on these experiments. Terpene and isoprene yields in CMAQ were reported by Carlton et al. (2011), which were based on experiments reported by Griffin et al. (1999). Griffin et al. (1999) performed their experiments in 20 m3 Teflon chambers in

Seinfeld's group at Caltech. Chhabra et al. (2011) conducted their experiment in 28 m3 Teflon chambers at Caltech, which are also operated by Seinfeld's group. Considering the factor that the chambers are similar in size and construction, and are operated by the same research group, using the correction factors derived by Chhabra et al. (2011) can be justified, lacking of reliable source of information. While the values used in this study can be debated, they are based on the best information we have currently. In the revised manuscript, the following paragraph is included:

"No changes of the saturation concentrations (C*) of the products were made although the mass yields were updated. This partial correction only represents an upper bound estimation of the wall-loss effect. It should also be noted that while the correction factors proposed for aromatic compounds by Zhang et al. (2014) were based on the same chamber experiments that the CMAQ yields were based on, the correction factors for alpha-pinene and isoprene reported by Zhang et al. (2014) were based on a different set of chamber experiments. This could lead to additional uncertainties in the yields used in this study."

15. Page 22, Table 4, Simulation 6: Clarify that anthropogenic VOC, NOx, SO2, etc were removed (not just VOC, NOX) Responses: Yes, all anthropogenic emissions were removed in simulation 6. A footnote is included to clarify this.

16. What is the major driver for how anthropogenic emissions affect SOA? Is it through POA? Responses: We indicated that the major driver is due to "reduced oxidation capacity of the atmosphere (i.e. lower OH during the day and NO3 radical at night) that leads to slower formation of semi-volatile and oligomers as well as lower acidity of the aerosols that reduces the uptake coefficient of IEPOX thus less SOA." (page 15, lines 10-13).

References

Lonneman, W. A.: Overview of VOC measurement technology in the PAMS program [microform], Washington, D.C.: Environmental Protection Agency, EP 1.23/6:600/A-

94/193, 1994. Zhang, H., and Ying, Q.: Secondary organic aerosol from polycyclic aromatic hydrocarbons in Southeast Texas, Atmospheric Environment, 55, 279-287, 10.1016/j.atmosenv.2012.03.043, 2012. Zhao, J., Levitt, N. P., Zhang, R., and Chen, J.: Heterogeneous Reactions of Methylglyoxal in Acidic Media:  Implications for Secondary Organic Aerosol Formation, Environmental Science & Technology, 40, 7682-7687, 10.1021/es060610k, 2006.
* * *